# Electrocrystallization of Calcium Oxalate on Electrospun PCL Fibers Loaded with Phytic Acid as a Template

**DOI:** 10.3390/polym14153190

**Published:** 2022-08-05

**Authors:** Tatiana Zegers Arce, Mehrdad Yazdani-Pedram, Andrónico Neira-Carrillo

**Affiliations:** 1Department of Biological and Animal Sciences, Faculty of Veterinary and Animal Sciences, University of Chile, Santiago, Santa Rosa 11735, La Pintana, Santiago 8820808, Chile; 2Department of Organic and Physical Chemistry, University of Chile, Olivos 1007, Independencia, Santiago 8380544, Chile

**Keywords:** calcium oxalate, polycaprolactone, phytic acid, electrocrystallization, polymer fibers

## Abstract

Crystallization occurs widely in living organisms where different organs could associate with the calcification process, such as the formation of calcium oxalate (CaOx) calculi in the urinary tract. However, the pathogenesis and the role of an inhibitor in the pathological processes involved in urolithiasis is poorly understood. Therefore, the use of phytic acid (PA) as an inhibitor for the organic fibrillar matrix is a novel approach to inhibit the formation of pathological CaOx crystals. Herein, electrospun polymer fiber meshes of polycaprolactone (PCL) with random (R) and aligned (A) fiber orientations containing PA were prepared by electrospinning, and their role as a 3D organic template in in vitro CaOx crystallization was investigated. CaOx crystals were generated on conductive tin indium oxide (ITO)-modified glass with R-PCL and A-PCL fibers in the presence of PA through an electrocrystallization (EC) procedure. This study provides a simple electrochemical approach to evaluate the role of PA as an inhibitor in the nucleation of pathological CaOx crystals. The resulting CaOx crystals were analyzed by chrono-potentiometry, optical microscopy (OM), scanning electron microscopy (SEM), and X-ray diffraction (XRD). We found that PA and the fiber orientations are key factors in the nucleation and crystal growth of CaOx, achieving the stabilization of healthy CaOx crystal and the inhibition of the pathological crystal form.

## 1. Introduction

Biological crystallization, often called ‘biomineralization’, is the process by which living organisms—ranging from bacteria to more complex forms of life—produce calcium-based biominerals (Ca-minerals) through an intimate association between inorganic and organic components under ambient conditions [1,2,3,4]. Calcium is the most abundant element in carbonates and oxalate biominerals [5]. The organic part of biominerals acts as an active template role controlling the nucleation, structure, morphology, crystal orientation and spatial confinement of the inorganic part. This differentiates biominerals from minerals. Biominerals present complex shapes, hierarchical organizations, and novel morphological structures with high strength properties and supporting functions [6,7]. Biominerals are highly organized from the molecular to the nano- and macroscale, with intricate architectures that ultimately make up a myriad of different functional soft and hard tissues [8].

Calcium oxalate (CaOx) fulfils various functions in living organisms; however, its precipitation is related to a pathological mineralization [9]. CaOx (CaC_2_O_4_·*n*H_2_O, where *n =* 1 to 3) exists mainly in three hydrated crystalline forms: monohydrated (COM, or whewellite), dihydrated (COD or weddellite) or trihydrated (COT or caoxite). COM is involved in kidney stones in the urinary tract in animals [10]. However, COM crystals are not attributed exclusively to pathology, as they can also be found in healthy individuals.

It is well known that the stereochemical and geometrical match between the chemical functional groups of organic components and ions in the organic phase dictates the orientation of crystals. Therefore, in vitro crystallization performed in the presence of anionic polymers or by additives in solution has been demonstrated to lead to the formation of polymorphism of minerals, where a wide range of organic templates has been used in inorganic mineralization to fabricate hierarchical hybrid materials. The precise control of the interfacial molecular recognition between the organic and inorganic parts of minerals and the stabilization of a particular hydrated form of CaOx have not been fully understood until today. The presence of a high level of oxalate ions and their capability to bind to Ca^2+^ ions can be associated to urolithiasis for susceptible persons, and can also be found in plants, where their importance in the food industry and biomedicine is increasing. Therefore, the determination of oxalic acid and/or a urinary oxalate sample utilizing oxalate-rich food sources as biosensor has been developed [11,12]. Indeed, proteins or polysaccharides present in urine or an additive such as citrate have the ability to modulate the crystallization process because they contain anionic groups that electrostatically interact with specific crystalline phases preventing the pathological formation of kidney stones [13,14,15]. For instance, many developments in the control of crystallization processes—along with their feasibility and implementation in industrial processes with abundant applications—have been investigated, and currently different advances in the crystallization procedure are being reported [16,17,18,19,20,21,22].

Therefore, the use of substances derived from plant sources as natural urolithiasis modulators of CaOx nucleation such as phytic acid (PA) or sarsaparilla molecules represents an interesting approach to the investigation of the pathological in vitro crystallization of minerals. Protective evidence of the phytate molecule of CaOx associated to nephrolithiasis was initially proposed by Modlin, and later the inhibition of CaOx crystallization using phytate was reported by Saw [23,24,25].

On the other hand, the topographic, chemical composition and three-dimensional (3D) fibrillar rearrangement properties of organic templates are important factors in the mineralization process [26]. In this regard, the effect of electrospun polymer mesh (ESM) on CaOx crystallization utilizing chitosan has been recently reported by us [27]. The formation of ESM has been studied intensively for several applications due to its ability to produce ultrafine fibers with controlled micro- or nanometric sizes and high specific surface areas [28]. 

We believe that ESM containing an additive molecule represents an excellent organic 3D matrix for in vitro inorganic mineralization. Indeed, ESM can possess a surface decorated with anionic chemical groups for mineralization, where the large surface area provides an electrostatic interaction with divalent Ca^2+^ ions inducing local ion concentration or anchoring additive molecules for active site groups. Electrospun polymer meshes (ESM) of polycaprolactone (PCL) with random (R-PCL) and aligned (A-PCL) fiber orientations were used as an organic 3D template to induce the formation of CaOx crystals by the electrocrystallization (EC) technique. Herein, the effect of random and aligned PCL-ESM loaded with PA on the in vitro CaOx EC and the inhibition effect of PA on the crystal growth or its specific capability to stabilize hydrated crystalline forms of CaOx is tested.

## 2. Materials and Methods

### 2.1. Reagents and Materials

Reagents of the highest available grade were utilized. Ultra-pure water (18.2 MΩ) was obtained from the LaboStarTM 4-DI/UV water system (LabostarTM TWF, Evoqua Water Technologies LLC, Warrendale, PA, USA). Solvents for the preparation of the EC solutions composed of sodium oxalate from Sigma-Aldrich (St. Louis, MO, USA), calcium nitrate tetrahydrate from Merck (Darmstadt, Germany), and ethylenediaminetetraacetic acid tetrasodium salt from Sigma-Aldrich were purchased. Polycaprolactone (80.000 mol wt) from Sigma-Aldrich and ITO substrate (10 × 25 mm) from Corning^®^ ITO glass (25 × 25 mm^2^; 1.1 mm-thick) from Delta Technologies (Dallas, TX, USA) were used. 

### 2.2. Preparation of the Electrospun PCL Fibers 

ESMs of controlled topology of PCL fibers (PCL-ESM) were directly deposited onto indium tin oxide (ITO) glasses which were used as templates for the in vitro EC of CaOx. For the preparation of the PCL-ESM fibers, 18% PCL (Mw, 80.000, Sigma-Aldrich) solution in ethyl acetone/acetate 3:1 (*v*/*v*) was utilized, which was kept on a magnetic stirrer at 40 °C until complete dissolution and left overnight at room temperature. Then, the PCL solution was placed in a 10 mL Nipro^®^ luer lock syringe. All of the PCL-ESMs were conducted in an eStretching LE-10 Fluidnatek^®^ instrument. 

Pieces of ITO substrates were glued onto the rotating drum and flat-plate collectors, and were further utilized as a working electrode for the EC procedure (Figure 1). For all of the working solutions, Milli-Q water was utilized. For the preparation of the random and aligned PCL fibers on the collectors, flat metal (30 × 30 cm^2^ at 15 cm) and rotation (10 cm in diameter at 2000 rpm for the collector) were used, respectively. The applied voltage and flow rate solution used for the preparation of both controlled PCL-ESM topologies were of 9.5 kV and 1000 µl/h, respectively. For the deposition of the PCL fibers, small cut pieces of ITO substrate (10 × 25 mm) from commercial Corning^®^ ITO glass were used. The ITO glasses have a sheet surface resistivity in the range of 5–15 Ω/sq.

### 2.3. Electrocrystallization of CaOx 

The CaOx EC on the ITO substrate contained random (R-PCL) or aligned (A-PCL) PCL fibers loaded with PA, and was performed in galvanostat/potentiostat (BASi Epsilon) (West Lafayette, IN, USA) equipment, as reported elsewhere [13,27]. Pieces of ITO glass were placed on a rotating drum and flat-plate collectors and directly modified with R-PCL or A-PCL fibers during the ES process, and then acted as a template on the in vitro EC technique. For the assays of EC, the ITO glass was immersed in an electrochemical cell containing a mixture of calcium nitrate, ethylenediaminetetraacetic acid (EDTA), and sodium oxalate solutions with a pH of 10.5. The EC of the CaOx was performed using 9 mA for 60 min, and the chronopotentiometry curves during the EC were recorded with a 2 s recording interval of sampling. The CaOx EC proceeds due to EDTA molecules bonded to Ca^2+^ ion inducing the Ca-EDTA complex in an alkaline pH medium, such that when an electric potential is applied, the electrolysis of water occurs, generating free O_2_ and protons in the vicinity of the ITO surface, with a consequent decrease in pH value and loss of Ca-EDTA complex stability [29]. Then, free Ca^2+^ ions react with oxalate ions, producing CaOx crystals which are subsequently deposited on the working electrode on ITO glass. The CaOx crystals’ formation on ITO generates the increases of its resistance during the passage of current, thereby increasing the recorded voltage during the CaOx EC.

### 2.4. Characterization of PCL-ESM and CaOx Crystals

CaOx crystals produced through EC on ITO were characterized by Fourier transform infrared spectroscopy (FTIR/ATR) in an Interspec 200-X^®^ (Interspectrum OU, Toravere, Estonia) instrument. The morphologies of the CaOx and PCL-ESM samples were observed by optical microscopy (OM) and scanning electron microscopy (SEM) in a Nikon Eclipse E400^®^ with the morphometric LAZ program (Image Pro-Plus, Media Cybernetics, Melville, NY, USA) and in a JEOL JSM-IT300LV microscope (JEOL USA Inc., Peabody, MA, USA), respectively. For the SEM analysis, PCL-ESM samples were gold-sputter-coated to thickness of 200 nm using a Denton Vacuum Desk V sputtering system in an argon atmosphere to render them electrically conductive. SEM observation was performed using an accelerating voltage of 20 kV. Moreover, the X-ray diffraction (PXRD) (Siemens, Munich, Germany) of PCL-ESM was conducted using a Siemens D-5000X X-ray diffractometer with CuKα radiation (graphite monochromator) and an ENRAF Nonius FR 590 X-ray generator. The crystal structure of the CaOx was determined using CuKα radiation (40 kV), a step scan of 0.2°, and the geometric Bragg–Brentano (θ-θ) scanning mode with an angle (2θ) in the range of 5–60°. The DiffracPlus program was used as the data control software.

### 2.5. Statistical Analysis

All of the data are expressed as an average ± standard deviation from at least three replicates. Statistical analysis was performed using differences between experimental groups considering an ordinary one-way analysis of variance by using the Minitab 19 program. In all cases, *p* < 0.05 was considered statistically significant.

## 3. Results

### 3.1. Chronopotentiometry

Figure 2 shows the chronopotentiometry behavior of the EC of CaOx performed on the bare ITO (control, Figure 2a) and in the presence of PCL-ESM meshes (A-PCL Figure 2b, and A-PCL Figure 2c). The inorganic mineral produced on the ITO surface is not a conductive material; therefore, its local formation reduces the electrode’s active area. Then, the resultant CaOx crystals constitute a barrier for oxygen diffusion, and in this way the total cathodic current decreases, which can be followed by fluctuations of the potential (V) measurements [30]. The registered current can be analyzed during the time of EC experiments in order to identify the deposited crystals when the electrode surface is covered. In general, Figure 2 shows that the recorded potential (V) behavior on bare ITO (control) and on ITOs modified with all PCL-ESM during the CaOx EC assays, in which the chronopotentiometry curve of CaOx EC was not similar and reached exclusively higher potential (V) values for A-PCL fibers than R-PCL fibers, demonstrating that the PCL have an active effect on the CaOx nucleation depending on the fiber orientations. The EC voltage in the control (Figure 2a) begins at 1.03 V, and its potential registration does not change upon reaching at the end of experiment at 1.01 V; however, the potential (V) registered on ITO glass modified with both PCL-ESM showed potential fluctuations associated with the formed CaOx crystals in the range of 1–8 min (Figure 2b,c), with the potential (V) being higher when the ITO was covered with A-PCL fibers at 1.4 V than R-PCL fibers at 1.15 V, respectively. 

We observed a slight increase with a noticeable rise at 3 min reaching a maximum of 1.42 V (Figure 2b), and 1.20 V (Figure 2c) at 8 min for A-PCL and R-PCL fibers, respectively. We also observed that EC essays with aligned and randomly ESM-PCL fibers with 1 mg/mL of PA (Figure 2c,d) and 1.5 mg/mL (Figure 2f,g) show a notorious difference from the potential (V) at the beginning of the EC of CaOx. In the case of A-PCL fibers with PA additive, both EC experiments started at the same potential (V) value close to 1.5 V (Figure 2d,f) and reached a higher potential (V) at higher concentrations of PA at 1.52 V and 1.45 V, respectively. When the CaOx EC was conducted with R-PCL fibers, the chronopotentiometry started at different potential values, being higher at 1.19 V for 1.5 mg/mL of PA and 1.03 V for 1.0 mg/mL. For the highest concentration of PA, the potential (V) value increased throughout the EC process associated with the nucleation and crystal growth of CaOx, reaching a value of 1.29 V (Figure 2g).

### 3.2. Optical Microscopy

Optical microscopy (OM) easily allows us to demonstrate the presence of CaOx on the A-PCL and R-PCL fibers supported on ITO glass at both PA concentrations (Figure 3). We observed different CaOx crystal morphologies in each of the EC essays by controlling the fiber orientation of PCL-ESM. Figure 3a shows the OM images of CaOx crystals obtained on the ITO substrate (control, without PCL fiber), on ITO with A-PCL fibers (Figure 3b), on ITO with R-PCL (Figure 3c), and on ITO with PCL-ESM using 1 and 1.5 mg/mL PA (A-PCL Figure 3d,e, and R-PCL Figure 3f,g), respectively. It is important to highlight that although we used a low-resolution technique, it was possible to notice and distinguish the different crystal morphologies and the flower-like crystal arrangement exclusively associated with PCL-ESM matrices and PCL mesh loaded with PA on ITO substrate.

In Figure 3a, the size of the CaOx crystals on the ITO surface are in the range of 10–40 µm, where flower-like COD crystals of size ca. 10 µm and a few irregular circular crystals are clearly observed. Figure 3b,c shows the formation of CaOx crystals on aligned and random PCL fibers. Here, a great abundance of crystals with similar morphologies was also distinguished. The presence of PCL-ESM on the ITO substrate does not inhibit the nucleation of CaOx formed on and inside PCL-ESM meshes. In the case of Figure 3d,e using aligned PCL-ESM and Figure 3f,g using random PCL-ESM with both PA concentrations, smaller CaOx crystals are associated with the formation of fibers mostly on their surfaces. Here, more defined flower like-crystals and circular CaOx crystals were observed. Generally, we found abundant crystals with similar morphologies in the absence and presence of PCL-ESM. 

### 3.3. Scanning Electron Microscopy

Scanning electron microscopy (SEM) allowed us to observe, in more detail, the morphological aspects of CaOx obtained by EC essays on the ITO substrate (Figure 4a), PCL-ESM (Figure 4b,c), and PCL-ESM with a PA additive (Figure 4d,g). The SEM images were consistent with previous OM observations, where the CaOx crystals obtained on ITO substrates, as well as on all PCL-ESM, showed a variability of sizes and shapes. The size of the CaOx crystals on the ITO varied between 5 and 30 µm, where flower-like COD crystals of a size ca. 20 µm and a few irregular circular crystals were observed (Figure 4a). On the other hand, Figure 4b,c shows that, for CaOx crystals obtained on A-PCL and R-PCL fibers, most of the crystals were distributed mainly on the PCL fibers, acting as a nucleation guide support. This was more evident when A-PCL fibers were used (Figure 4d,e). Indeed, in some images, the fibers seem to be crossed by the COD crystals, as indicated with blue arrows in Figure 4e and g. From the data in Figure 4g, it is apparent that abundant and defined bipyramidal COD crystals on the A-PCL template with 1.5 mg/mL PA were obtained. In the case of random PCL-ESM with 1 and 1.5 mg/mL PA (Figure 4f,g), similar COD and circular crystals were observed. We also observed more abundant spherical CaOx crystals at a higher PA concentration. The presence of PCL-ESM does not disturb the formation of CaOx obtained on and inside of PCL-ESM meshes.

Additionally, the SEM-EDS of CaOx crystals obtained in all of the in vitro EC assays was carried out with PCL-ESM (Figure 5b,c) and PCL-ESM with PA (Figure 5d,e). EDS measurements (Figure 5a–c) showed the presence of the typical constitutive elements of CaOx crystals, such as calcium (Ca), carbon (C) and oxygen (O) atoms (Figure 5a). Additionally, due to the sensitivity of the EDS microanalysis and the penetrability of the X-ray beam, other elements such as indium (In), aluminum (Al) and silicon (Si) atoms from the ITO glass substrate were also detected. In fact, when a PA additive was utilized at a higher concentration on the R-PCL and A-PCL meshes in the CaOx EC assays (Figure 5d,e), not only were the typical elements from CaOx and PCL-ESM detected but also the phosphorus (P) atom from the PA additive, being uniformly distributed throughout the PCL-ESM sample. 

### 3.4. X-ray Diffraction (XRD)

X-ray diffraction analysis was used to characterize the CaOx phases formed in all of the CaOx EC assays on the ITO substrate using PCL-ESM samples in the presence of PA (Figure 6). Additionally, the XRD pattern of the ITO glass and both PCL fiber matrices are shown in the Appendix A, in which the crystallographic peaks corresponding to ITO glass, A-PCL and R-PCL fibers are clearly distinguished. In general, the XRD spectra of CaOx crystals obtained on ITO substrate (Figure 6a) for PCL-ESM (Figure 6b,c), A-PCL (Figure 6d,e) and R-PCL (Figure 6f,g) meshes with 1 mg/mL and 1.5 mg/mL PA showed diffraction peaks assigned to COD and COM phases, as well as to ITO and PCL, respectively. In Figure 6a, the XRD of CaOx on ITO shows crystallographic peaks at (2 Theta degree) 2θ = 14.9°, 24.3°, and 37.7°, which can be correlated to the (hkl) indices (−101), (020), (311) of the COM phase and four extra peaks belonging to the ITO substrate at 2θ = 21.4°, 30.3°, 35.4°, 45.5° and 50.7° (Appendix A). 

Figure 6b,c shows a similar XRD pattern for CaOx obtained on ITO with PCL fibers without a PA additive in which the crystallographic peaks of ITO are still present but new peaks belonging to COD and COM phases with different intensities appear, showing a coexistence of both hydrated crystalline forms. The crystallographic peaks at 2θ = 14.3° (D), 20.1° (D), 23.8°(M), 24.3° (M), 32.2° (D), 37.4° (D), 47.9° (D) were observed corresponding to the (200), (211), (−211), (020), (411), (103) and (413) planes, respectively. When the CaOx EC essays were performed in the presence of an A-PCL fiber using both 1 mg/mL and 1.5 mg/mL PA, fewer crystallographic peaks associated to COM and more intensive peaks of COD were found (Figure 6d,e). The XRD of CaOx formed on R-PCL and A-PCL meshes and using the two PA concentrations demonstrated that they controlled the CaOx crystallization processes, promoting the formation of COD crystals. Indeed, when the CaOx EC was carried out with R-PCL fiber at both PA concentrations, all of the crystallographic peaks corresponding to COM showed signals with less intensity (Figure 6f,g); surprisingly, we also we found that the crystallographic peak at 2θ = 14.9° (M) associated to the pathogenic (−101) plane of COM disappeared when the R-PCL with 1 mg/mL of PA was utilized (Figure 6f).

The unidentified peaks corresponding to some hydration state of CaOx were assigned by using the XRD patterns of JCPDS card N°: 20–231 and 17–541 as COM and COD, respectively. 

On the other hand, the Minitab 19 program (Appendix A) was used to determine the number of CaOx crystals, estimation of the average number of CaOx crystals (number of crystals/mm^2^), confidence interval obtained in the fibrillar matrices according to the type and orientation of the fibers, additive concentration, and determination of the different statistical considerations in order to ascertain the average sizes of the CaOx crystals, and to compare pairs and differences between the different EC tests. From the analysis of the average crystal size, using the normality test (Anderson–Darling) and the test for equality of variances (Bonferroni–Levene), the results showed that there was a non-normal distribution (Appendix A) and heteroscedasticity, as the variance of errors was not constant (Appendix A). Considering the experimental nature of the CaOx EC assays, where the samples are independent, and given the non-normality, heteroscedasticity and independence of the data, the determination of the difference between the mean crystal sizes was carried out using Welch’s Test; once the statistical difference was confirmed, the comparison in pairs was carried out by the Games Howell method (Appendix A). The estimation of the confidence intervals and the determination of the differences between the different EC tests for CaOx and their comparisons are shown in Table 1, Table 2, Table 3, Table 4, Table 5 and Table 6.

The Games Howell method showed that there was a statistical difference in the average sizes of CaOx crystals between CN-A (control test) and the EC tests in the presence of A-PCL at two concentrations of PA. The effect of PA on the variation of the average sizes of CaOx crystals demonstrated that smaller and larger crystal sizes were obtained when 1 mg/L and 1.5 mg/L PA in the EC procedure were utilized. The CaOx crystals obtained with A-PCL mesh without PA are of intermediate size. 

The Games Howell method revealed a significant statistical difference in the n values obtained between the control test (CN-R) and the CaOx EC tests using random PCL mesh with PA concentrations of 1.5 and 1. mg/mL. The effect of PA on the variation of n was demonstrated, resulting in smaller CaOx crystals when PA was used at a concentration of 1 mg/L, and larger crystals when the EC was performed with R-PCL without additive PA. Crystals with an intermediate n value were obtained when using the R-PCL mesh with PA at 1.5 mg/L. A comparison between the groups (Table 1 and Table 2) allows the quantitative estimation of the effect of the PA additive at its different concentrations and for the types of mesh in the EC tests, considering the evaluation of one factor at a time in the analysis of the results. Further analysis by making comparisons within groups (Table 3 and Table 4) allows us to simultaneously evaluate the effect of PA at the two concentrations according to the type of PCL mesh and the crystals obtained in the EC tests of CaOx.

The Games Howell method showed that there was a statistical difference in the n values obtained between CN-A and PA-A at 1.5 mg/L; however, there was no statistical difference when the PA additive was used at 1 mg/L. Additionally, when using PA at 1 mg/L, the crystals apparently had a lower n value compared to CN-A; however, the Games Howell method showed that there was no statistical difference. This analysis demonstrates that the largest average crystal sizes in these EC assays were achieved with the PA-A mesh using PA at 1.5 mg/L. This indicates that the effect of increasing the concentration of the PA additive under the experimental conditions was significant.

The Games Howell method showed that there was a statistical difference in the n values between the control trial (CN-R) with PA-R with both concentrations in all of the EC assays. It was observed that there was a statistical difference in the values of n in each EC test when using 1 mg/L and 1.5 mg/L PA. It was shown that the smallest-size crystals are achieved with 1 mg/L PA, and that the largest crystals are obtained only with the R-PCL mesh, followed by the EC test with the PA additive at 1.5 mg/L. Again, it was shown that the effect of increasing the concentration of PA in the experimental conditions of the EC was statically significant in the results of n obtained.

Table 5 shows that the total comparison in all of the EC tests and analyzed by Games Howell pairwise comparison in a single comparison to evaluate the effect of PA in EC assays at both concentrations is independent of the type of mesh utilized.

The Games Howell method showed that there is a statistical difference between the median of the crystal sizes—that is, carrying out EC assays of CaOx on different types of mesh and with the utilized PA concentrations—is significant, which reflects its effect on the variation of the n values, finding that the largest and smallest crystalline sizes were obtained with the A-PCL and R-PCL meshes using PA 1.5 mg/mL and 1 mg/mL, respectively.

In addition, the results obtained in all of the EC assays of CaOx were analyzed in the ordinal data manner, according to the total number (N) and average size (n) of the crystals, in order to ascertain the effect of different experimental conditions (Table 6). This data analysis provided information about the relationship between the PA with a particular experimental condition, to understand better how the PA additive modulated the N and n of crystals compared to control EC assay, and about the effect of PA on the organic template in the pathological CaOx mineralization involved in urolithiasis. In general, regarding the N values, it was observed that the CN-R control assay and the use of PA-A at 1.5 mg/mL achieve the lowest positions 1 and 2, respectively: on the other hand, the PA-A at 1 mg/mL presented the highest N value, placing it in the highest position.

When comparing the CE assays using the aligned mesh, the control (CN-A) is in an intermediate position (fourth place) according to N, and third place according to the n value. If we compare it with the PA-A 1.5 mg/L assay, it presents fewer crystals (second place); however, according to the crystal size, it presented the largest average crystal size (seventh place). Furthermore, the EC assay with PA-A 1 mg/L showed the highest total number of crystals (seventh place), and ranked second with respect to its average size, indicating its improved positioning. The EC assays with PA-R and PA-A at 1.5 and 1 mg/L showed a large increase in the number of crystals (fifth and sixth places), respectively; however, a decrease in the size of their crystals was also observed (fourth and second places), respectively. The foregoing demonstrates a very radical variation between the different EC assays. As indicated in Table 5, there was a significant difference between these and the control, such that both options would be an improved experimental condition compared to the negative control (BC) using the aligned CNO mesh. 

On the other hand, when comparing the EC assays on random mesh, the negative control (CN-R) presented the lowest number of total crystals (N); however, it presented the second highest value of the average sizes (sixth place). This was followed by EC assays with PA-R of 1 mg/L (third place); however, the size of the crystals (n) was considerably reduced (first place), and the PA-R of 1.5 mg/L (fifth place) with average crystalline sizes (n) had the position of fourth place. Therefore, this analysis again confirms that if there were significant differences between the two EC assays, both options would also be an improved experimental condition compared to the control (CN-R) mesh. 

## 4. Discussion 

OM analysis helped us to preliminarily demonstrate the presence of CaOx crystals in all of the in vitro CaOx EC assays. However, SEM analysis allowed us to resolve the individual morphologies of COD and COM with more structural detail, and to demonstrate the A-PCL and R-PCL fiber distribution supported on the ITO substrate. The SEM-EDS microanalysis allowed us to identify the Ca, C and O atoms as constitutive elements of the CaOx crystals; the PCL-ESM meshes; the In, Si and Al elements of ITO glass; and the P atom of the PA additive. The XRD analysis of CaOx showed the presence of only COM in the control EC assays, whereas a predominance of COD crystals was found when the ITO was modified with both PCL-ESM meshes in the negative controls. The EC assays with R-PCL mesh showed a higher proportion of the COD phase compared to COM. We found that when EC assays of CaOx was performed with 1 mg/mL PA on both PCL-ESM meshes, a higher proportion of COD was observed. In particular, in the EC using R-PCL with PA at 1 mg/mL, only the healthy COD crystal was observed, and the most interesting finding was an absence of the crystallographic plane (−101) associated to the pathological crystal form of COM. The chronopotentiometry revealed an increase of the potential (V) value when the concentration of PA increased during the initial deposition of crystals onto the ITO surface. We believe that phosphorous groups of PA molecules interact with Ca^2+^ ions, forming a PA-PO_3_H-Ca complex that serves as source of mineral precursor deposition, which subsequently incorporates oxalate ions, forming the final form of CaOx. The CaOx crystal deposition increases the resistance on the ITO surface, hindering the passage of the applied current, thereby increasing the potential (V), as was observed for PCL-ESM at both PA concentrations. The potential curve during the entire period of the EC assay reflects a complex equilibrium of a free Ca^2+^ diffusion, mineral dissolution and PCL-ESM solubilization at the interface of the ITO surface. Therefore, the phosphate groups of the PA molecule can strongly interact with Ca^2+^ ions to provide nucleation sites, thus attracting the anionic oxalate groups on the PCL meshes, giving rise to a heterogeneous nucleation, inhibiting the nucleation of COM and promoting the formation of COD crystals. On the other hand, chronopotentiometry showed variations in the voltage recorded during the EC assays, in which a distinctive potentiometric curve for each type of PCL-ESM mesh was registered. It was observed that the A-PCL mesh had higher initial potential (V) than the R- PCL in the absence and presence of additive PA. We also found a concentration-dependent relationship, where the chronopotentiometry of EC assays with A-PCL and R-PCL at PA 1.5 mg/mL showed higher V values in each PCL-ESM system used. These results are consistent with the effect of both PCL-ESM systems at the two PA concentrations (Table 6). The stabilizing effect of the COD form was demonstrated by SEM-EDS and XRD, even for lower concentrations of PA.

## 5. Conclusions

Controlled-topology PCL-ESMs loaded with a PA additive on an ITO glass substrate were prepared and used as a solid template for the EC of CaOx, and their effects on the morphology and crystal growth of CaOx were analyzed by MO, SEM-EDS, chronopotentiometry and XRD. PA had a modulating effect on the EC of CaOx, showing an effect of reducing the size (n) of CaOx, promoting the metastable COD phase with a bipyramidal morphology, inhibiting the COM phase, and achieving the absence of the (−101) plane associated to the pathological form of COM. The effect of PCL-ESM containing PA on the CaOx EC can probably be explained by the presence of anionic phosphorus groups of the PA additive that influence the nucleation and crystal growth of COD crystals. It was statistically revealed that the most effective EC experimental condition for controlling CaOx nucleation was achieved by using R-PCL at 1 mg/mL PA, which reached a significant reduction in the size (n) of the CaOx, stabilizing only the COD form. The chronopotentiometry showed that PA had a concentration-dependent kinetic inhibitory effect on the EC of CaOx. We believe that by utilizing this simple and efficient electrochemical crystallization associated with the use of 3D electrospun polymer fiber loaded with anionic molecules provides a viable in vitro Ca-mineral crystallization procedure for the study of various aspects of biomineralization, which includes crystal production with controlled morphologies and the possibility to study the inhibition or promotion process associated with urolithiasis based on CaOx, which is a common pathological mineralization and is often a subject of client consultation in veterinary clinics. Finally, the EC provides a precise and simple in vitro crystallization assay with controllable experimental parameters of EC set up in which the electrical current, time, concentration of reactants, and type of additive allows us to control the morphology and type of CaOx crystals.

## Figures and Tables

**Figure 1 polymers-14-03190-f001:**
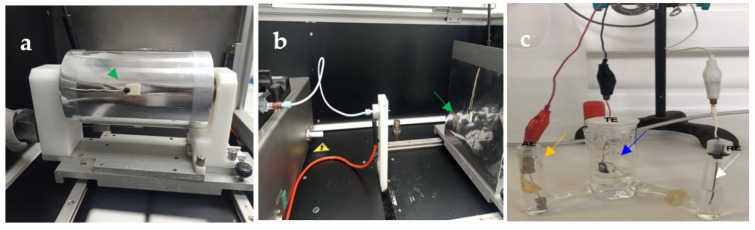
Optical images illustrating the manufacture of PCL–ESM on pieces of ITO glass utilized as a working electrode on the CaOx EC glued to (**a**) a rotating drum (rotation speed of 1800 rpm, clockwise), (**b**) a flat plate collector, and (**c**) the EC of the CaOx set-up. The green arrows indicate the working electrode connected to the ITO on both collectors covered with aluminum foil, and the blue, white and yellow arrows show the working, reference and auxiliary electrodes connected to ITO soaked in EC solution.

**Figure 2 polymers-14-03190-f002:**
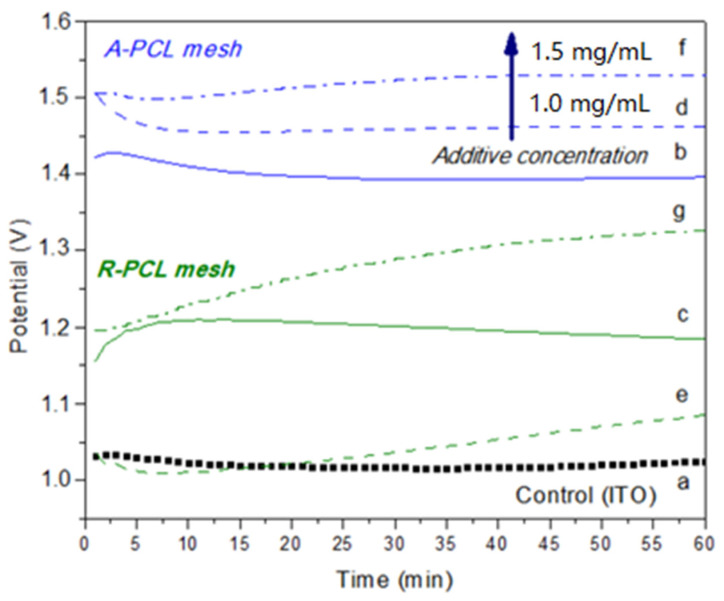
Chronopotentiometry of EC of CaOx performed on ITO modified with PCL fibers and PA at two concentrations. (**a**) Control (bare ITO), (**b**) A-PCL, (**c**) R-PCL, (**d**) A-PCL with 1 mg/mL PA, (**e**) R-PCL with 1 mg/mL PA, (**f**) A-PCL with 1.5 mg/mL PA, and (**g**) R-PCL with 1.5 mg/mL PA.

**Figure 3 polymers-14-03190-f003:**
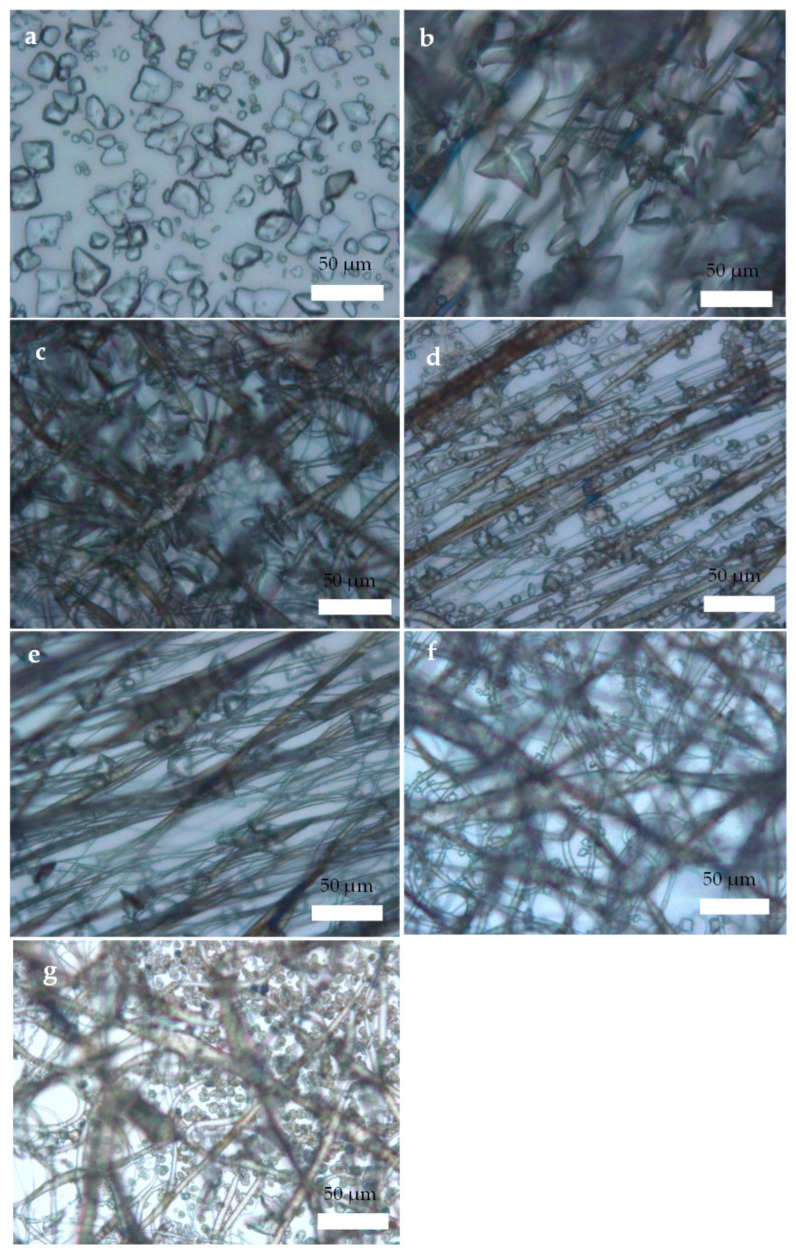
OM images of CaOx crystals obtained via EC on ITO modified with PCL fibers: (**a**) control (without PCL fibers), (**b**) A-PCL, (**c**) R-PCL, (**d**) A-PCL with 1 mg/mL PA, (**e**) A-PCL with 1.5 mg/mL PA, (**f**) R-PCL with 1 mg/mL PA, and (**g**) R-PCL with 1.5 mg/mL PA. (40× magnifications).

**Figure 4 polymers-14-03190-f004:**
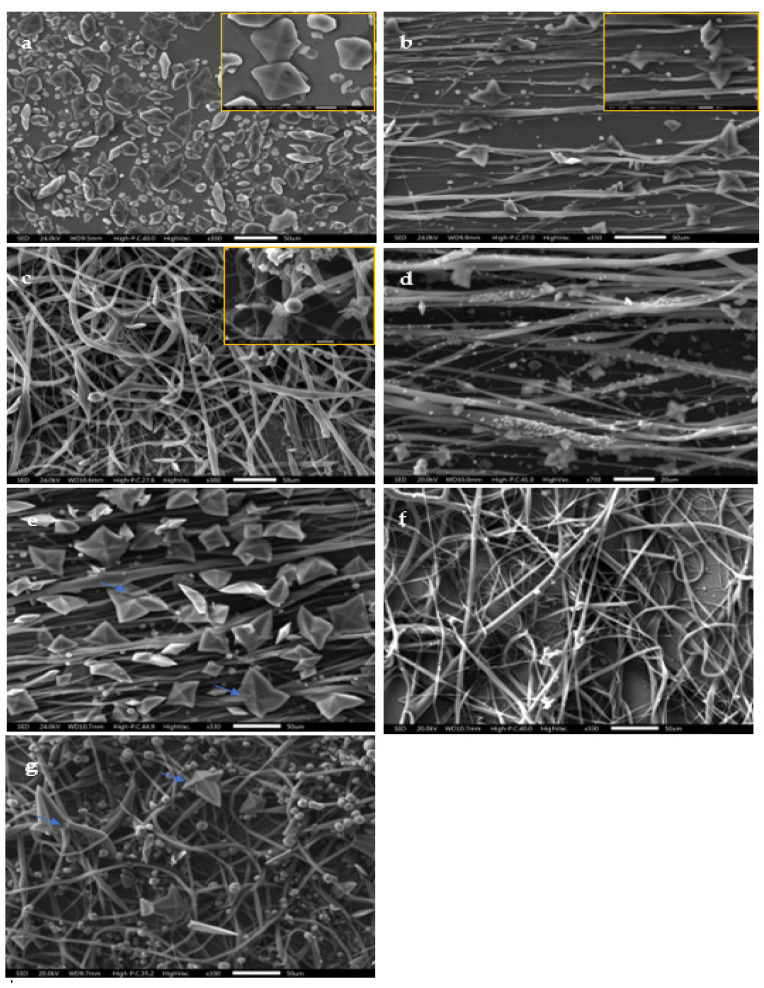
SEM images of CaOx crystals obtained via EC on ITO modified with PCL fibers: (**a**) control (without PCL fibers), (**b**) A-PCL, (**c**) R-PCL, (**d**) A-PCL with 1 mg/mL PA, (**e**) A-PCL with 1.5 mg/mL PA, (**f**) R-PCL with 1 mg/mL PA, and (**g**) R-PCL with 1.5 mg/mL PA (40× magnifications).

**Figure 5 polymers-14-03190-f005:**
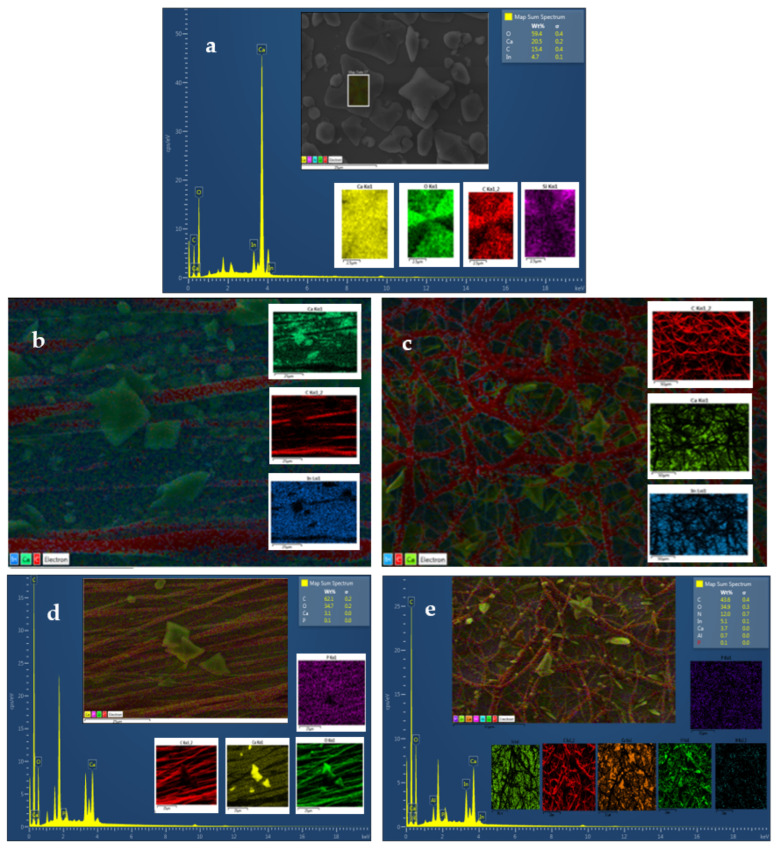
SEM-EDS images of CaOx crystals obtained via EC on ITO modified with PCL fibers: (**a**) control (ITO without PCL fibers), (**b**) A-PCL, (**c**) R-PCL, (**d**) A-PCL with 1.5 mg/mL PA, and (**e**) R-PCL with 1.5 mg/mL PA.

**Figure 6 polymers-14-03190-f006:**
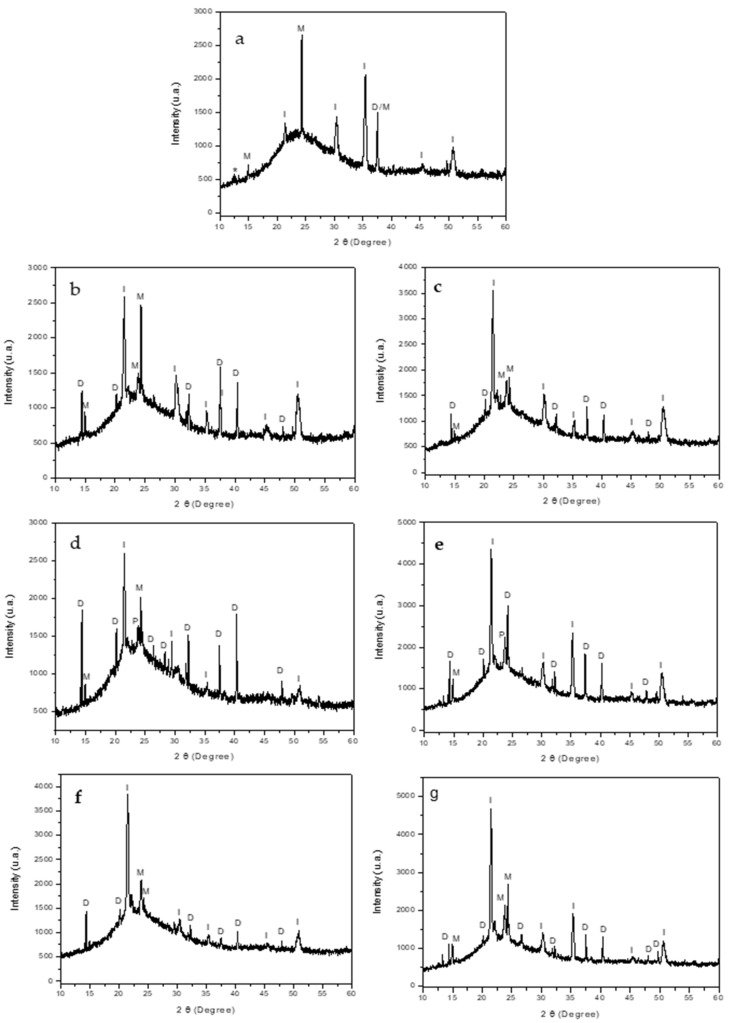
XRD diffractograms of the CaOx crystals obtained via EC on ITO modified with PCL fibers: (**a**) control (without PCL fibers), (**b**) A-PCL, (**c**) R-PCL, (**d**) A-PCL with 1 mg/mL PA, (**e**) A-PCL with 1.5 mg/mL PA, (**f**) R-PCL with 1 mg/mL PA, and (**g**) R-PCL with 1.5 mg/mL PA. The designations of D, M, I, P and * correspond to reflections from COD, COM, ITO, and PCL fibers, and the unidentified phases, respectively.

**Table 1 polymers-14-03190-t001:** Games Howell pairwise comparison between the negative control assay using A-PCL and its corresponding CaOx EC assays on this mesh with two PA concentrations.

Factor	n	Median	Grouping		
PA-A 1.5 mg/L	99	20.64	A		
CN-A	108	7.933		B	
PA-A 1 mg/L	208	5.598			C

The control trial is indicated in gray, and the results of the smallest and largest average sizes are shown in green and red, respectively. The median values that do not share a letter indicate that they are significantly different.

**Table 2 polymers-14-03190-t002:** Games Howell pairwise comparison between the negative control assay using R-PCL and its corresponding EC assays on this mesh with two PA concentrations.

Factor	n	Median	Grouping		
CN-R	38	15.70	A		
PA-R 1.5 mg/L	161	9.578		B	
PA-R 1 mg/L	109	4.445			C

The control assay is indicated in gray, and the results of the smallest and largest average sizes are shown in green and red, respectively. The median values that do not share a letter indicate that they are significantly different.

**Table 3 polymers-14-03190-t003:** Games Howell paired comparison between the negative control assay using A-PCL and its corresponding EC assays on this mesh with two PA concentrations.

Factor	n	Median	Grouping		
PA-A 1.5 mg/L	99	20.64	A		
CN-A	108	7.933		B	
PA-A 1 mg/L	208	5.598		B	

The control assay is indicated in gray, and the results of the smallest and largest average sizes are shown in black and red, respectively. The median values that do not share a letter indicate that they are significantly different.

**Table 4 polymers-14-03190-t004:** Games Howell pairwise comparison between the negative control assay using R-PCL and its corresponding EC assays on this mesh with two PA concentrations.

Factor	n	Median	Grouping		
CN-R	38	15.70	A		
PA-R 1.5 mg/L	161	9.578		B	
PA-R 1 mg/L	109	4.445			C

The control assay is indicated in gray, and the results for the smallest and largest average sizes are shown in green and red, respectively. The median values that do not share a letter indicate that they are significantly different.

**Table 5 polymers-14-03190-t005:** Games Howell pairwise comparison between the negative control assays using the A-PCL and R-PCL meshes, and the corresponding EC assays on these meshes at two PA concentrations.

Factor	n	Median	Grouping						
PA-A 1.5 mg/L	99	20.64	A						
CN-R	38	15.70	A	B					
BC	172	12.965		B					
PA-R 1.5 mg/L	161	9.578							
CN-A	108	7.933			C	D	E		
PA-A 1 mg/L	208	5.598					E	F	
PA-R 1 mg/L	109	4.445						F	G

The control assay is indicated in gray, and the results for the smallest and largest average sizes are shown in green and red, respectively. The median values that do not share a letter indicate that they are significantly different.

**Table 6 polymers-14-03190-t006:** Ordinal comparison considering the total number of crystals and average crystal size in each CaOx EC assay, in the control assay in the absence of mesh (BC), in the presence of CN-A) and CN-R meshes, and in the presence of PA-A and PA-R meshes at 1mg/L, and 1.5 mg/L, respectively.

	Total Number of Crystals (N)	Average Crystal Size (n)
CN-R	1°	6°
PA-A 1.5 mg/L	2°	7°
PA-R 1 mg/L	3°	1°
CN-A	4°	3°
PA-R 1.5 mg/L	5°	4°
BC	6°	5°
PA-A 1 mg/L	7°	2°

The 1° number corresponds to the lowest value of N and n in each EC assay of CaOx.

## Data Availability

The data presented in this study are available on request from authors T. Zegers Arce, M. Yazdani-Pedram and A. Neira-Carrillo.

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
