# Peer review of "Electrocrystallization of Calcium Oxalate on Electrospun PCL Fibers Loaded with Phytic Acid as a Template"

_polymers, 2022, doi:10.3390/polym14153190_

Round 1

Reviewer 1 Report

In this study authors studied the growth of Pathological crystallization of CaOx crystals into organic fibrillar matrices, and the mechanism was investigated carefully. Then, Electrospun polymer fiber meshes of polycaprolactone with random and aligned fiber orientation by electrospinning were performed. The resulting CaOx crystals obtained by EC were analyzed by differen analysis, which demonstrated that PA additive and the fiber orientations are key factors in the nucleation and crystal growth of the non-pathological crystal type of CaOx. In my opinion, this work is attractive and can be published in this journal after some correction:

1-The novelty aspect of this study should be highlighted in abstract section

2-Authors should be discussed about the effects of PCL in the texts

3-It is necessary to propose a related mechanism for better understanding of the efficiency of material.

4-Recent advances on crystallization procedure are advised included, e.g.
A) doi.org/10.1016/j.matpr.2021.05.211, B) doi.org/10.1016/j.jallcom.2022.165404, C:doi.org/10.1016/j.molliq.2022.118676, D: doi.org/10.22052/JNS.2021.02.014, E: doi.org/10.1590/fst.37821, F: doi.org/10.1155/2021/3250058

5- English writing should be improved by native person

Reviewer 2 Report

Review

Manuscript Number: polymers-1834714

Title: Electrocrystallization of Calcium Oxalate on the Electrospun PCLFibers Loaded with Phytic Acid as Template

In this paper, the authors have examined the growth mechanisms of CaOx crystals into organic fibrillar matrices. They have applied electrospun polymer fiber meshes of polycaprolactone (PCL) with random (R) and aligned (A) fiber orientation and analysed the active role of phytic acid (PA) on in vitro crystallization in the hydrated crystalline forms of CaOx. It was found that that PA additive and the fiber orientations are key factors in the nucleation and crystal growth of the non-pathological crystal type of CaOx.

There are some questions and remarks to be answered:

1.    The authors should perform a proof reading of the text (some mistakes, typos, etc.).

2.    There is a lack of graphical abstract.

3.    There is no information on the applied reagents (chemical purity, producer, etc).

4.    Authors should compare the efficiency of proposed method with conventional methods.

5.        The authors should rewrite Conclusions, considering the comparison with literature data (proving that this method is better than others).

6.        Authors should supplement References with a larger number of literature items.

Reviewer 3 Report

The current manuscript provides an interesting account of electrocrystallization of Calcium Oxalate on the Electrospun PCL Fibers. However, there are some comments for the manuscript that the authors need to address:

1. The application part of the manuscript and the research is not clear. It is partially mentioned in the introduction but the end use is not provided.

2. The authors mentioned that "Indeed, the ESM can surface decorated with anionic chemical groups for mineralization, where the large surface area provides an electrostatic interaction with divalent Ca2+ ions inducing local ion concentration or anchoring additive molecules for active site groups". I do not find information in the manuscript confirming the above aspect.

3. Information in Figure 5 is not clear and it is not translated well in the text.
